# Dynamic Changes in Drinking Behaviour among Subpopulations of Youth during the COVID-19 Pandemic: A Prospective Cohort Study

**DOI:** 10.3390/healthcare11131945

**Published:** 2023-07-05

**Authors:** Mahmood R. Gohari, Thepikaa Varatharajan, James MacKillop, Scott T. Leatherdale

**Affiliations:** 1School of Public Health Sciences, University of Waterloo, 200 University Avenue West, Waterloo, ON N2L 3G1, Canada; t8varath@uwaterloo.ca (T.V.); sleatherdale@uwaterloo.ca (S.T.L.); 2Peter Boris Centre for Addictions Research, St. Joseph’s Healthcare and McMaster University, 100 West 5th Street, Hamilton, ON L8P 3R2, Canada; jmackill@mcmaster.ca

**Keywords:** secondary school students, developmental trajectory, binge drinking, latent transition analysis, quasi-experiment

## Abstract

Objective: Youth drinking is highly heterogenous, and subpopulations representing different alcohol use patterns may have responded differently to the COVID-19 pandemic. This study examined changing patterns of alcohol use in subpopulations of the youth population over the first two years of the pandemic. Method: We used linked survey data from 5367 Canadian secondary school students who participated in three consecutive waves of the COMPASS study between 2018/19 and 2020/21. Latent transition analysis (LTA) was used to identify patterns of alcohol use based on the frequency of drinking and frequency of binge drinking and to estimate the probability of transitioning between identified patterns. Results: LTA identified five patterns of alcohol use each representing a unique subpopulation: abstainer, occasional drinker-no binging, occasional binge drinker, monthly binge drinker, weekly binge drinker. Probability of being engaged in binge drinking for a subpopulation of occasional drinkers pre-pandemic was 61%, which reduced to 43% during the early-pandemic period. A lower proportion of occasional binge drinkers reported moving to monthly or weekly binge drinking. Female occasional drinkers were more likely to move to binge drinking patterns during the pandemic than males. Conclusions: Less frequent drinking and younger students were more likely to reduce their drinking and binge drinking than more established drinkers during the COVID-19 pandemic. Understanding of heterogenous patterns of alcohol drinking and different responses to public health crises may inform future preventive programs tailored to target subpopulations more effectively.

## 1. Introduction

The COVID-19 pandemic has profoundly impacted the lives of young people in various aspects. To curb the transmission of the virus, governments worldwide implemented stringent public health measures, including emergency lockdown protocols and stay-at-home orders. These measures have significantly disrupted the daily routines and social lives of many young individuals, resulting in extended periods of confinement at home with their families and limited opportunities for interaction with friends, peers, and community networks [1]. This unique situation has the potential to influence changes in the alcohol consumption patterns among youth. On one hand, some individuals may turn to increased alcohol consumption as a coping mechanism in response to the stresses and challenges brought about by the pandemic. On the other hand, reduced opportunities for social drinking contexts and limited access to alcohol-related environments could lead to a decrease in alcohol consumption among young people during this period [2]. Longitudinal studies investigating the impact of COVID-19 on youth drinking and binge drinking behaviours have found conflicting results thus far. While some studies have indicated increases in the frequency and quantity of alcohol consumption since the pandemic started [3,4,5,6], others have reported a decrease [7,8,9,10]. Despite these studies exploring changes in youth alcohol consumption during the pandemic, very little is known about the changes in alcohol use behaviours among subgroups of youth. Instead of addressing the possibility that youth vary in their patterns of use as they age and could be clustered into distinct subgroups, the primary focus of existing COVID studies has been predicting the direction and magnitude of a single growth trajectory for alcohol and/or binge drinking [3,4,8]. However, previous longitudinal studies conducted both before and during the pandemic have recognized that the population exhibits heterogeneity in terms of drinking behaviors [11,12,13,14]. These studies have identified that patterns of alcohol use vary among youth in terms of the timing and intensity of the transitions [15,16,17]. Since subgroups could also respond differently to the pandemic, grouping youth based on their drinking patterns may help to identify those groups that have been disproportionately affected by the pandemic. For instance, as previously indicated for alcohol and cannabis [18,19], we may expect that in response to social event restrictions during the pandemic, occasional drinkers are probably more likely to change their frequency of drinking compared to regular drinkers, who have a more established drinking behaviour are more likely to maintain their consumption. By targeting interventions towards specific subgroups and addressing their unique challenges and risk factors, we can enhance the effectiveness of prevention efforts and promote healthier behaviours among young individuals.

The current study examined changing patterns of alcohol consumption in subpopulations of the youth population over the pandemic period. Using data from an ongoing prospective cohort study of Canadian secondary school students, our study employed latent transition analysis (LTA) to examine the profiles of drinking behaviour among subpopulations of the youth population. LTA, an extension of latent class analysis, allows for the investigation of heterogenous changes in longitudinal data by identifying distinct groups of individuals based on their responses to multiple questions [20]. We employed two key indicators of drinking behaviours, the frequency of alcohol use and the frequency of binge drinking, to measure the multidimensional behaviour of alcohol consumption to provide a comprehensive understanding of the behaviour, in contrast to previous research that primarily relied on a single alcohol use indicator [7,21]. This approach allowed us to predict which subgroups were more likely to experience changes in their drinking behaviour and in what direction those changes would occur. Drawing from this knowledge, this paper explores the impact of the pandemic by comparing transition probabilities between subgroups of the population before and after the pandemic. Given that there are sex differences in alcohol use behaviours and in the different mechanisms that are being used to respond to the pandemic, a secondary purpose of this paper was to consider changes in males’ and females’ subpopulations separately.

## 2. Material and Method

The study utilized data from the COMPASS (Cannabis use, Obesity, Mental health, Physical activity, Alcohol use, Smoking, and Sedentary behaviour) study. COMPASS is an ongoing annual data collection initiative that involves a sample of Canadian students in grades 9–12 (secondary I–V in Quebec). The study recruited participants from a large convenience sample of secondary schools. Students were eligible to take part in the study if their parents provided active-information and passive-consent through a parental permission protocol [22].

### 2.1. Design and Participants

Longitudinal changes were examined by analyzing three-year linked data from students who participated in the study during the 2018/19 (T1), 2019/20 (T2), and 2020/21 (T3) academic years. The data was collected from a total of 81 schools located in Quebec (*n* = 37), Ontario (*n* = 38), and British Columbia (*n* = 6). During the initial data collection at T1, students completed a paper-based survey that was administered to whole-school samples within a single classroom period [23]. In T2, data collection was conducted pre-pandemic (from September 2019 to February 2020, T2a) and post-pandemic (from May to June 2020, T2b), creating two cohorts to disaggregate changes in alcohol use due to age-related factors from period-related factors. T2a data were collected as in T1. As the COVID-pandemic was declared in Canada in March 2020 and schools were closed to in-person learning, T2b data were collected online using the Qualtrics XM survey software (Qualtrics, Provo, UT, USA). Online data collection continued for both cohorts in T3 given the ongoing pandemic response.

An anonymous unique identification code was created for students using information from five questions asked at the beginning of the questionnaire to link student data across study waves [24]. As shown in Figure 1, data were linked between three years to construct two cohorts: Cohort T2a includes students who provided linked data between T1, T2a, and T3 (*n* = 3467) while Cohort T2b provided linked data between T1, T2b, and T3 (*n* = 1900). Figure 1 indicates that both cohorts participated in all three waves of data collection, and the only difference between them is that Cohort T2a only includes students surveyed before the onset of the pandemic in T2a, while Cohort T2b only includes students surveyed after the onset of the pandemic in T2b.

### 2.2. Measures

The study utilized two self-reported items to assess alcohol consumption among participants, drawing on items that have been previously used in national youth substance use surveillance [25]. To capture the frequency of drinking alcohol, participants were asked the following question: “In the last 12 months, how often did you have a drink of alcohol that was more than just a sip”? Based on the distribution of responses, the frequency of drinking was categorized into the following groups: no drink, less than once a month (occasional drink), once to three times a month (monthly drink), and once a week or more (weekly drink).

The frequency of binge drinking, defined as consuming 5 or more drinks on a single occasion, was assessed using the question: “In the last 12 months, how often did you have 5 drinks of alcohol or more on one occasion”? Similar to the coding for alcohol consumption, the responses for binge drinking were categorized as follows: no binge drink, less than once a month (occasional binge drink), once to three times a month (monthly binge drink), and once a week or more (weekly binge drink).

## 3. Analyses

We employed a three-stage analytical approach to investigate the patterns of alcohol use behaviour among the participants. In the first stage, we utilized Latent Transition Analysis (LTA) to identify the optimal number of latent alcohol use patterns. Two variables were used to measure the frequency of drinking and the frequency of binge drinking, each with four levels. We tested the fit of 2 to 6 status models across the three waves of data. To ensure robustness, each model was fit with 50 different sets of random starting values for the parameters, allowing us to obtain the best maximum likelihood estimates [26]. The models’ relative fits were compared using several fit indices, including AIC and adjusted BIC. Additionally, we assessed the classification quality of the model within the two cohorts by examining the average posterior probabilities of latent status membership. Higher posterior probabilities indicate better classification of the different statuses [20].

In the second stage, we examined the stability of the underlying structure of alcohol use behaviour across the three waves for the two cohorts. We compared the fit of a constrained LTA model, where the item-response probabilities (IRPs) were constrained to be identical over the three waves, with an unconstrained LTA model where the parameters were estimated freely. This comparison allowed us to assess whether the latent structure remained stable or changed over the course of the study.

Third, we examined the impact of the pandemic by comparing fit of a model with and without cohorts as a covariate, where the cohort is defined 1 (if the cohort was Cohort 2b) and 0 (if the cohort was Cohort 2a). We also explored probabilities of transitioning to latent statuses from pre-pandemic (T1) to early-pandemic (T2) and from early-pandemic (T2) to ongoing-pandemic (T3).

We utilized the Procedure LTA (Latent Transition Analysis) in SAS [27] to identify latent statuses and estimate transition probabilities among these statuses. This analytical tool allowed us to explore the dynamics of transitions within the latent statuses over time. This procedure handles incomplete outcome variables using a full maximum likelihood approach, which assumes that missing data are missing at random [28]. It incorporates missing covariates in the analysis by maximizing the likelihood function with respect to both observed and missing data. To capture any gender-specific patterns or differences in alcohol use behaviour, we conducted stratified models separately for males and females.

To examine attrition, we examined whether participant characteristics differed between those that were included in the study compared to those that were excluded because their data were not linked over the three years. Results of the attrition analysis indicated that slightly more students in the analytic sample identified as being female and white, and with lower levels of drinking and binge drinking than the excluded participants (See Appendix A).

In terms of missing data, of the total 16,101 records from 5367 participants, the proportion of records with complete data is 96.6% (*n* = 15,549), and 3.4% (*n* = 552) records had missing, whereas 246 had only one, 289 had two, and 20 had more than two missing values. The rates of missing drinking were 0.8% (pre-pandemic), 2.5% (early-pandemic), and 5.2% (ongoing-pandemic) and of missing binge drinking were 0.2% (pre-pandemic), 0.2% (early-pandemic), and 5.4% (ongoing-pandemic). For other missing patterns, we used a pairwise deletion approach [29].

## 4. Results

### 4.1. Preliminary Analyses

Table 1 shows characteristics of the two cohorts. The proportion of students for each race and age group was similar between both cohorts, but Cohort T2b comprises of more female than Cohort T2a (57.3% vs. 65.2%). Figure 2 indicates frequency of drinking and binge drinking by cohorts at baseline and over three years. It appears that very similar proportions of the two cohorts were engaged in different levels of drinking and binge drinking at baseline. The proportions of drinking levels remained the same between the two cohorts over the two years of the pandemic; however, Cohort T2b students reported slightly higher levels of binge drinking than Cohort T2a students during the two years of follow-up (Figure 2b).

### 4.2. Latent Subpopulation Analyses

The results of statistical fit indices of 2 to 6 latent class models (Appendix A) signifies that adjusted BICs of the models decreased from 2 to 5 status models, after which they began to increase at the 6 status model. Based on the fit indices and the conceptual relevance of the latent classes, we selected the 5 status model as our final model. This model produced clearly distinct and interpretable patterns of youth alcohol use. The posterior probabilities of latent status membership for the T2a cohort were 0.91, 0.89, and 0.86 in the pre-, early-, and ongoing-pandemic periods, respectively. For the T2b cohort, the probabilities were 0.94, 0.91, and 0.88. These values indicate a high quality of classification within the two cohorts.

The item response probabilities presented in Table 2 characterize five latent statuses, which we have labeled as: abstainer (had never consumed alcohol or did not consume alcohol in past year), occasional drinker-no binging (reported once a month or less drinking and no binge drinking), occasional binge drinker (once a month or less drinking and binge drinking), monthly binge drinker (less than 3 times drinking and binge drinking in a month), weekly binge drinker (more than 3 times a week drinking and binge drinking). Before examining the possibility of changes in the patterns of use between the two cohorts, we examined the stability of underlying structures over the three years. The results of LTA indicated that the latent status structure and interpretation were stable over the three waves of the study in both Cohort T2a. ($G_{restericted\;model}^{2}-G_{unrestericted\;model}^{2}=856.3-795.8=60.5,\;\Delta df=4021-3961=60,\;p=0.476)$ and Cohort T2b ($G_{restericted\;model}^{2}-G_{unrestericted\;model}^{2}=1074.1-1153.1=79.0,\;\Delta df=60,\;p=0.052)$.

### 4.3. Cohort Differences in Latent Subpopulation Analyses

After examining the stability of classes over the three-year study period, Figure 3a shows changes in the size of the five latent statuses between the two cohorts. From pre-pandemic to early-pandemic, the proportion of abstainers that maintained their abstinence was similar in both Cohort T2a and Cohort T2b. The escalation in the proportion of Cohort T2b students who were engaged in occasional binge drinking (from 7% to 10%) was lower than the escalation among the Cohort T2a students (from 8% to 16%). The increase in the proportions of monthly binge drinkers more than doubled among Cohort T2a students (from 6% to 15%), while it only increased 1.5 times (from 9% to 13%) among Cohort T2b students. In contrast, the size of weekly drinking and binge drinking in Cohort T2b increased slightly more than that of Cohort T2a. It appears that changes in the size of latent statuses of the two cohorts from T2 to T3 were similar (Figure 3b).

To investigate the potential impact of the pandemic on the likelihood of transitioning between different alcohol use statuses from pre-pandemic to early-pandemic and from early-pandemic to ongoing-pandemic, we conducted a comparison of the 5-status model with and without a cohort covariate. The inclusion of the cohort variable did not yield a significant result in the model (*p* = 0.276). This suggests that, while students in Cohort T2b reported lower levels of drinking and binge drinking during the early-pandemic wave compared to Cohort T2a students, the overall influence of the pandemic on the probability of transitioning between different alcohol consumption statuses was not statistically significant.

While the overall difference between the two cohorts was not significant, changes in subpopulations, identified by each drinking status, seem different. To explore switching in and out of different drinking statuses that led to changes in the size of latent statuses, Table 3 compares the two cohorts in terms of transition probabilities between the latent statuses during the study period. From pre-pandemic to early-pandemic, abstainers in Cohort T2b had a 71% chance of remaining abstinent at the early-pandemic stage, a chance that was slightly higher than that of Cohort T2a with 67%. Abstainers in Cohort T2b had also slightly lower chance of occasionally binge drinking at the early-pandemic stage relative to Cohort T2a (4% vs. 6%). Findings suggests that Cohort T2a occasional drinkers-no binging had a 37% chance of being engaged in occasional binge drinking, 22% chance of monthly binge drinking, and 2% weekly binge drinking. In other words, the chance of being engaged in binge drinking for occasional drinkers of Cohort T2a was 61% during the pre-pandemic stage, which reduced to 42% during the early-pandemic stage. The likelihood of reducing the frequency of binge drinking and moving from occasional binge drinking to occasional drinking more than doubled during the early-pandemic stage compared to the pre-pandemic, where students in Cohort T2b reported 30% chance of reduction relative to Cohort T2a students who had only 12% chance.

### 4.4. Sex Differences

We examined sex difference between transitions in latent drinking statuses by including sex as a covariate. The result showed that sex was significantly related to status membership (*p* < 0.001). To explore the transition probabilities in females and males, we fit two LTA models to male and female data separately. Fit indices support the 5-status solution for both females and males (Appendix A). Transition probabilities (Appendix A) suggest that from T1 to T2, overall, males had lower escalation in binge drinking than females. Male occasional drinkers-no binging in Cohort T2b had more than double the chance of maintaining non binge drinking than Cohort T2a (87% vs. 40%), the chance that was 1.5-fold for females (52% vs. 35%).

## 5. Discussion

This study examined the changes in alcohol-use behaviours among two cohorts of underage secondary school students before and during the COVID-19 pandemic, within the framework of a natural experiment. The unique contribution of this research lies in its investigation of the dynamics of alcohol use within specific subgroups of the population, going beyond the overall impact on the entire population. We initially identified five alcohol use patterns characterised as abstainer, occasional drinker-no binging, occasional binge drinker, monthly binge drinker, and weekly binge drinker. While overall results suggest that there is no evidence on early impact of the pandemic on youth drinking, our more advanced analyses showed that subgroups of the population did have very different responses to the pandemic. During the early-pandemic stage, occasional drinkers-no binging were less likely to escalate their binge drinking as they did during the pre-pandemic stage, while more established drinkers continued drinking as pre-pandemic. Youth in both cohorts indicated similar probabilities of escalation in their alcohol use from early- to ongoing-pandemic stages as both have been exposed to the pandemic. Females were more likely to escalate binge drinking than males.

Contrary to the literature that suggests a rise in alcohol use and binge drinking during adolescence [30], our results showed that the escalation rate among Cohort T2b students was lower than Cohort T2a students. This finding supports the Availability Theory [2], which posits reduced youth alcohol consumption during the pandemic is a consequence of reduced opportunities to drink. For example, social restrictions imposed by the Canadian Government in response to the pandemic may have inadvertently lowered rates of escalation and/or reduction in the frequency of binge drinking as youth are no longer able to socialize with peers in settings that involve alcohol (i.e., social events or parties) [1,31]. Furthermore, spending more time at home in close proximity to their parents/guardians may restrict access to alcohol (i.e., no longer can rely on older peers as a source).

The stratified analyses identified some unique sex differences. During the early-pandemic period, female students experienced higher elevations in their binge drinking than males. These findings could be partially explained by the notion that females engage in binge drinking to help alleviate their mental health problems, such as stress and depression [32,33], whereas males’ binge drinking is driven by a desire to match the drinking habits of their peers [34]. 

We also observed that during the initial year of the pandemic, there was a more significant reduction in the increase of drinking and binge drinking frequencies among participants who engaged in occasional drinking compared to those who consumed alcohol more frequently. Similar results were found in studies focusing on alcohol and other substances (e.g., cannabis), where youth were more likely to decrease, discontinue, or had slower rates of increasing consumption if they were less-frequent users [18,19]. Our results, which show that three of the four drinking classes were driven by frequency of binge drinking, also support the notion that binge drinking is the most common pattern of alcohol use among youth and can be differentiated based one’s level of risk [33]. Future prevention efforts should consider targeting high-risk binge drinking behaviours. By focusing on high-risk binge drinking behaviours, prevention programs can effectively target those individuals who are more vulnerable to the immediate and long-term harms associated with excessive alcohol consumption.

There are a few key features that distinguishes our study from others. First, with our longitudinal data we can assess the ongoing impacts of the COVID-19 (i.e., from pre-pandemic to two years during the pandemic), which is a limitation for the few available longitudinal studies that collect repeated data up to the first year of the pandemic. Furthermore, most studies evaluating the impacts of the COVID-19 pandemic utilize a cross-sectional design, which is prone to recall error due to participants having to retrospectively remember what their alcohol use was like from pre- to during the pandemic. Second, we are able to assess pandemic-related escalations of alcohol use (adjusting for known age-related increase) using our quasi-experimental design with a comparison group. Finally, by using a latent transition approach, our study is unique in that we can examine the effects of the pandemic on specific population subgroups rather than focusing on the trajectories of the overall population.

Limitations of the study should be considered when interpretating results. First, despite our large sample size, which is comprised of youth from a diverse range of communities, the sampling method employed by COMPASS study is not representative of all Canadian youth. Second, there is potential for participation bias with the COMPASS study switching from a paper-based questionnaire to an online questionnaire [35]. The effect of change in data collection mode may be confounded with pandemic effects. Third, youth may have misreported (e.g., underreported) their drinking as this study utilized self-reported data, which are subject to biases (i.e., recall or social desirability) [36]. However, it is unlikely that our results were influenced by measurement errors as (1) the applied LTA method assigned youth to latent statuses based on their posterior membership probabilities and (2) all cohorts had similar measurement errors [20]. Fourth, as shown in our attrition analysis, we may have presented a more conservative estimate for the prevalence of high-risk latent statuses. This is because our linked dataset is less likely to capture students who dropped out of school due to their substance use. Additionally, there could be a potential overlap in the data regarding the frequency of alcohol consumption between the pre-pandemic period and the early stages of the pandemic. This is particularly relevant for individuals who consume alcohol infrequently, such as those who drink less than once a month. Their change in drinking habits may have occurred prior to the pandemic, but they reported it during the early pandemic period, which does not necessarily reflect the impact of the pandemic itself. Finally, we recognize that the pandemic impacts are not singular but instead reflects ongoing and evolving effects.

## 6. Conclusions

This longitudinal study identified five youth population subgroups with respect to alcohol use behaviours. Our findings indicate that different subgroups of the population responded differently to the challenges posed by the pandemic. During the early and ongoing pandemic period, moving to binge drinking among low-frequent users was less likely than escalation in binge drinking of youth with more established drinking behaviours. This reduction in escalation of binge drinking might be because of less opportunities to binge drink during the pandemic. This observation suggests that future prevention programs targeting alcohol use may also need to be tailored to the specific needs and characteristics of the population subgroups. For certain subgroups that showed a pronounced decrease in alcohol consumption during the pandemic, policies could be designed to reinforce and sustain these positive changes. On the other hand, if other subgroups exhibited increased alcohol consumption, policies can be developed to address the underlying factors contributing to this trend, such as providing targeted support and resources to address mental health issues or social isolation.

## Figures and Tables

**Figure 1 healthcare-11-01945-f001:**
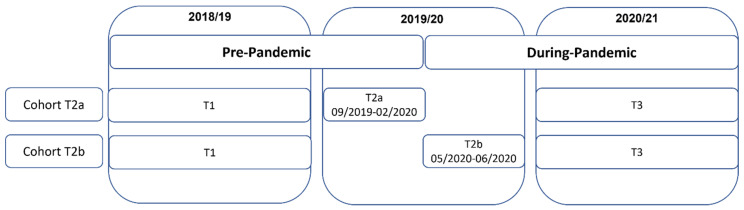
Data collection periods and longitudinal data linkage. Cohort T2a consists of students who provided linked data between T1, T2a, and T3 (*n* = 3467) and Cohort T2b consists of students who provided linked data between T1, T2b, and T3 (*n* = 1900).

**Figure 2 healthcare-11-01945-f002:**
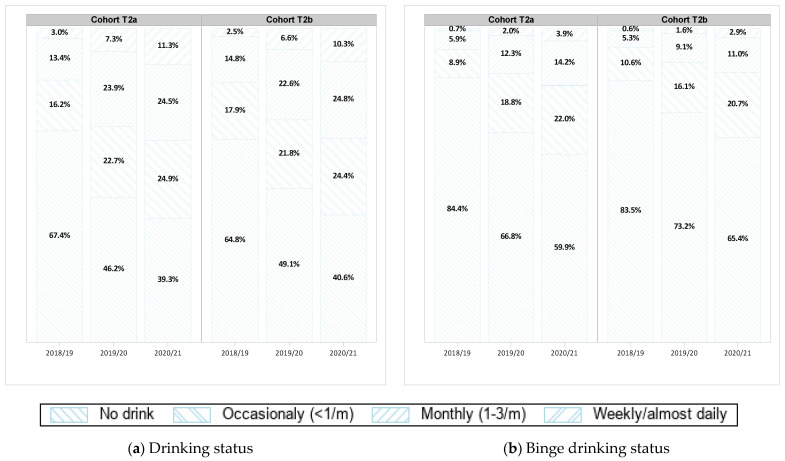
Proportions of different levels of overall (**a**) alcohol consumption and (**b**) binge drinking by cohorts over three years of COMPASS study from 2018/19 to 2020/21. Cohort T2a consists of youth who provided linked data in 2018/19, September 2019 to February 2020, and 2020/21. Cohort T2b consists of students who provided linked data in 2018/19, May 2020 to June 2020, and 2020/21.

**Figure 3 healthcare-11-01945-f003:**
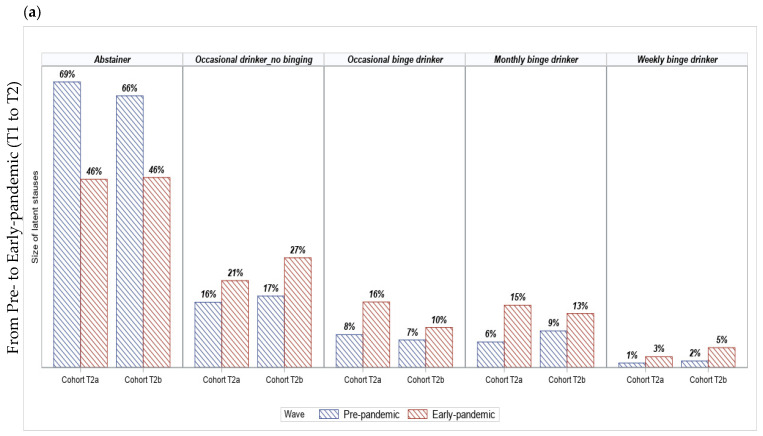
Changes in the size of drinking statuses from (**a**) pre-pandemic 2018/19 (T1) to early-pandemic 2019/20 (T2) and (**b**) from early-pandemic (T2) to ongoing-pandemic 2020/21 (T3) by cohorts. Cohort T2a consists of youth who provided linked data in 2018/19, September 2019 to February 2020, and 2020/21. Cohort T2b consists of students who provided linked data in 2018/19, May 2020 to June 2020, and 2020/21.

**Table 1 healthcare-11-01945-t001:** Distribution of baseline characteristics of the two cohorts of students attending the 81 linked-longitudinal COMPASS schools across the three study waves (2018/19, 2019/20, and 2020/21).

Demographics	Cohort T2a (*n* = 3467); *n* (%)	Cohort T2b (*n* = 1900); *n* (%)
Sex (female)	1987 (57.3)	1239 (65.2)
Age (years)		
12–13	1227 (35.5)	747 (39.5)
14	1341 (38.8)	603 (31.9)
15–18	885 (25.7)	542 (28.6)
Race (White)		
White	2755 (79.5)	1532 (80.6)
Black	47 (1.4)	48 (2.5)
Asian	231 (6.6)	101 (5.4)
Latino/Hispanic	37 (1.1)	17 (0.9)
Mixed/Others	397 (11.4)	202 (10.6)

Cohort T2a consists of youth who provided linked data in 2018/19, September 2019 to February 2020, and 2020/21. Cohort T2b consists of students who provided linked data in 2018/19, May 2020 to June 2020, and 2020/21. Due to missing data on characteristics, the number of subgroups may not add up to the overall sample.

**Table 2 healthcare-11-01945-t002:** Item-response probabilities in the identified structure of latent statuses of youth alcohol use behaviours in three waves (school year 2018/19 to 2020/21) of the COMPASS study (*n* = 5367).

	Abstainer	Occasional Drinker-No Binging	Occasional Binge Drinker	Monthly Binge Drinker	Weekly Binge Drinker
Frequency of drinking					
None	0.977	0.150	0.006	0.000	0.002
Occasional use	0.022	0.595	0.577	0.024	0.017
Monthly use	0.000	0.236	0.402	0.780	0.039
Weekly use	0.000	0.18	0.15	0.196	0.942
Frequency of binge drinking					
None	1.000	0.971	0.000	0.097	0.057
Occasional use	0.000	0.021	0.991	0.304	0.035
Monthly use	0.000	0.008	0.009	0.597	0.397
Weekly use	0.000	0.000	0.000	0.003	0.511

**Table 3 healthcare-11-01945-t003:** Comparison of transition probabilities between latent statuses of youth alcohol consumption among two cohorts of secondary school students attending the 81 linked-longitudinal COMPASS schools across the three study waves (2018/19, 2019/20, and 2020/21) from pre-pandemic to two years during the COVID-19 pandemic.

	Status Membership at Subsequent Year									
	Abstainer		Occasional Drinker-No Binging		Occasional Binge Drinker		Monthly Binge Drinker		Weekly Binge Drinker	
Status Membership at Baseline	Cohort T2a	Cohort T2b	Cohort T2a	Cohort T2b	Cohort T2a	Cohort T2b	Cohort T2a	Cohort T2b	Cohort T2a	Cohort T2b
**Pre-Pandemic (T1)**	T2									
Abstainer	**0.67 ***	**0.71**	0.20	0.20	0.06	0.04	0.06	0.04	0.01	0.01
Occasional drinker	0.01	0.01	**0.38**	**0.56**	0.37	0.21	0.22	0.21	0.02	0.01
Occasional binge drinker	0.03	0.01	0.12	0.30	**0.38**	**0.31**	0.43	0.33	0.04	0.04
Monthly binge drinker	0.02	0.05	0.03	0.06	0.09	0.03	**0.70**	**0.63**	0.16	0.22
Weekly binge drinker	0.03	0.01	0.06	0.06	0.03	0.01	0.18	0.18	**0.70**	**0.75**
**Early-Pandemic (T2)**	T3									
Abstainer	**0.76**	**0.79**	0.16	0.16	0.04	0.02	0.03	0.02	0.01	0.01
Occasional drinker	0.09	0.00	**0.49**	**0.59**	0.26	0.21	0.12	0.17	0.04	0.03
Occasional binge drinker	0.10	0.01	0.19	0.32	**0.43**	**0.46**	0.25	0.16	0.03	0.05
Monthly binge drinker	0.03	0.03	0.04	0.04	0.20	0.12	**0.54**	**0.62**	0.19	0.19
Weekly binge drinker	0.05	0.03	0.04	0.03	0.06	0.00	0.21	0.18	**0.64**	**0.76**

* Probabilities of maintaining behaviours are bolded; Cohort T2a consists of students who provided linked data in 2018/19, September 2019 to February 2020, and 2020/21 (*n* = 3447) and Cohort T2b consists of students who provided linked data in 2018/19, May 2020 to June 2020, and 2020/21 (*n* = 1900).

## Data Availability

Access to the COMPASS study data can be requested by completing and obtaining approval through an online form available at: https://uwaterloo.ca/compass-system/information-researchers/data-usage-application (accessed on 1 May 2023). For any inquiries regarding dataset access, please direct your requests to https://uwaterloo.ca/compass-system/ (accessed on 1 May 2023).

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
