# Peer review of "Dynamic Changes in Drinking Behaviour among Subpopulations of Youth during the COVID-19 Pandemic: A Prospective Cohort Study"

_healthcare, 2023, doi:10.3390/healthcare11131945_

Round 1
Author Response
Response to the reviewers' comments
Reviewer #1
Thank you for your careful review of our manuscript and the constructive comments.
Comment 1. In the following paragraph, page 7, the authors establish possible causes that promp different drinking behaviours in males and females. Two references are provided to back the possible influence of mental problems on drinking behaviour in females, whereas binge drinking in males would be more related to social norms. For this late statement, no support is provided. Please, revise the paragraph to avoid a simplistic interpretation of data.
Response. We revised the paragraph and added the reference to support our statement. The revised paragraph reads “The stratified analyses identified some unique sex difference. During the Early-pandemic period, female students experienced higher elevations in their binge drinking than males. These findings could be partially explained by the notion that females engage in binge drinking to help alleviate their mental health problems, such as stress and depression (Smith et al., 2021; Miller et al., 2007), whereas males' binge drinking is driven by a desire to match the drinking habits of their peers (Dir et al., 2017).”
Reference.
Dir, A. L., Bell, R. L., Adams, Z. W., & Hulvershorn, L. A. (2017). Gender differences in risk factors for adolescent binge drinking and implications for intervention and prevention. Frontiers in psychiatry, 8, 289.
Comment 2. The following paragraph (on page 5) should be revised (double comma and use of capital letters for the word cohort)
Response. Thank you, fixed!
Comment 3. Revise the following sentence on page
Response. The error in the name of the cohort is fixed now.
olation.

Reviewer 2 Report
The article is general well written. I just have a few minor comments to help improve the impact of the paper and clarity. My main comment relates to expanding on how the results of this study are useful now and in the future. I think this needs to be discussed.
Introduction
Minor comments:
1. The following section is unreferenced “For instance, we may expect that in response to social event restrictions during the pandemic, occasional drinkers are probably more likely to change their frequency of drinking com[1]pared to regular drinkers, who have a more established drinking behaviour are more likely to maintain their consumption”. It would be good for the authors to provide some evidence as to why they predict this.
2. It would be good for the authors to further comment on the following: Since subgroups could also respond differently to the pandemic, grouping youth based on their drinking patterns may help to identify those groups that have been disproportionately affected by the pandemic. Some discussion on why this is important now (given the world is moving on from the pandemic) is vital. For example, how might identifying these groups be useful now or in the future?
Methods
1. More details about the validity and reliability of the measures used would improve the methods section of the manuscript.
Discussion
More discussion is needed on how this research can be applied to the world now? The world is moving on from lockdowns and the pandemic. What about the current results could be important for the future. What are some key implications (theoretical and clinical/practical) from the current result:
1. For example the authors state “future prevention efforts should consider targeting high-risk binge drinking behaviours” which could be extended on.
2. Also, “Understanding of heterogenous patterns of alcohol drinking and different responses to public health crises or policy changes may allow for future developments of preventive programs tailored to target subpopulation to deliver more effective programs” could be expanded upon. How might policies be changes or Reponses due to results of this study, how might programs be tailored?
Author Response
Response to the reviewers' comments
Reviewer #2
The article is general well written. I just have a few minor comments to help improve the impact of the paper and clarity. My main comment relates to expanding on how the results of this study are useful now and in the future. I think this needs to be discussed.
Response. Thank you for your careful review of our manuscript and the constructive comments. We acted on your comments and expanded on the policy and program implications of our findings in the discussion section as it has listed in our responses to comments below.
Introduction
Minor comments:
- The following section is unreferenced “For instance, we may expect that in response to social event restrictions during the pandemic, occasional drinkers are probably more likely to change their frequency of drinking com[1][1]pared to regular drinkers, who have a more established drinking behaviour are more likely to maintain their consumption”. It would be good for the authors to provide some evidence as to why they predict this.
Response. Thank you! We revised the sentence and it is read as “For instance, as previously indicated for alcohol and cannabis (Benchop et al., 2021 and Leatherdale et al, 2021), we may expect that in response to social event restrictions during the pandemic, occasional drinkers are probably more likely to change their frequency of drinking compared to regular drinkers, who have a more established drinking behaviour are more likely to maintain their consumption”.
Leatherdale, S. T., Brown, K. S., Carson, V., Childs, R. A., Dubin, J. A., Elliott, S. J., . . . Sabiston, C. M. (2014). The COMPASS study: a longitudinal hierarchical research platform for evaluating natural experiments related to changes in school-level programs, policies and built environment resources. BMC Public Health, 14(1), 1-7.
Benschop, A., et al. (2021). "Changing patterns of substance use during the coronavirus pandemic: Self-reported use of tobacco, alcohol, cannabis, and other drugs." Frontiers in psychiatry
- It would be good for the authors to further comment on the following: Since subgroups could also respond differently to the pandemic, grouping youth based on their drinking patterns may help to identify those groups that have been disproportionately affected by the pandemic. Some discussion on why this is important now (given the world is moving on from the pandemic) is vital. For example, how might identifying these groups be useful now or in the future?
Response. We expanded on the sentence by adding a sentence as “Since subgroups could also respond differently to the pandemic, grouping youth based on their drinking patterns may help to identify those groups that have been disproportionately affected by the pandemic. By targeting interventions towards specific subgroups and addressing their unique challenges and risk factors, we can enhance the effectiveness of prevention efforts and promote healthier behaviors among young individuals.”
Methods
- More details about the validity and reliability of the measures used would improve the methods section of the manuscript.
Response. Thank you! The two measures used to measure the frequency of alcohol drinking and binge drinking is similar to those validated measure used in the national study of Canadian Student Tobacco, alcohol, and Drug Survey and they are established measures of youth drinking. We referenced this national study in the paper.
Discussion
More discussion is needed on how this research can be applied to the world now? The world is moving on from lockdowns and the pandemic. What about the current results could be important for the future. What are some key implications (theoretical and clinical/practical) from the current result:
- For example the authors state “future prevention efforts should consider targeting high-risk binge drinking behaviours” which could be extended on.
Response. We revised the sentence and it reads as “future prevention efforts should consider targeting high-risk binge drinking behaviours. By focusing on high-risk binge drinking behaviors, prevention programs can effectively target those individuals who are more vulnerable to the immediate and long-term harms associated with excessive alcohol consumption.”
- Also, “Understanding of heterogenous patterns of alcohol drinking and different responses to public health crises or policy changes may allow for future developments of preventive programs tailored to target subpopulation to deliver more effective programs” could be expanded upon. How might policies be changes or Reponses due to results of this study, how might programs be tailored?
Response. We expanded on this sentence by adding a few specific policy and program implications of our finding of heterogeneous response patterns of subpopulation to the pandemic as follows. Understanding of heterogenous patterns of alcohol drinking and different responses to public health crises or policy changes may allow for future developments of preventive programs tailored to target subpopulation to deliver more effective programs. For example, if certain subgroups showed a pronounced decrease in alcohol consumption during the pandemic, policies could be designed to reinforce and sustain these positive changes. On the other hand, if other subgroups exhibited increased alcohol consumption, policies can be developed to address the underlying factors contributing to this trend, such as providing targeted support and resources to address mental health issues or social isolation.
